# Design and Implementation of Machine Tool Life Inspection System Based on Sound Sensing

**DOI:** 10.3390/s23010284

**Published:** 2022-12-27

**Authors:** Tsung-Hsien Liu, Jun-Zhe Chi, Bo-Lin Wu, Yee-Shao Chen, Chung-Hsun Huang, Yuan-Sun Chu

**Affiliations:** 1Communications Engineering Department, National Chung Cheng University, Chiayi 62102, Taiwan; 2Electrical Engineering Department, National Chung Cheng University, Chiayi 62102, Taiwan

**Keywords:** deep learning, DNN, ORM, speech enhancement, machine tools, life period

## Abstract

The main causes of damage to industrial machinery are aging, corrosion, and the wear of parts, which affect the accuracy of machinery and product precision. Identifying problems early and predicting the life cycle of a machine for early maintenance can avoid costly plant failures. Compared with other sensing and monitoring instruments, sound sensors are inexpensive, portable, and have less computational data. This paper proposed a machine tool life cycle model with noise reduction. The life cycle model uses Mel-Frequency Cepstral Coefficients (MFCC) to extract audio features. A Deep Neural Network (DNN) is used to understand the relationship between audio features and life cycle, and then determine the audio signal corresponding to the aging degree. The noise reduction model simulates the actual environment by adding noise and extracts features by Power Normalized Cepstral Coefficients (PNCC), and designs Mask as the DNN’s learning target to eliminate the effect of noise. The effect of the denoising model is improved by 6.8% under Short-Time Objective Intelligibility (STOI). There is a 3.9% improvement under Perceptual Evaluation of Speech Quality (PESQ). The life cycle model accuracy before denoising is 76%. After adding the noise reduction system, the accuracy of the life cycle model is increased to 80%.

## 1. Introduction

With the development of science and technology, a lot of manpower is replaced by automated equipment, and machine tools can process products more accurately. However, if the equipment is damaged, such as the milling cutter of the machine tool is worn or defective, and the maintenance personnel cannot replace the milling cutter in time, it will lead to the decline and loss of product quality. In the past, the more common maintenance method was reactive maintenance. When the machine is in an abnormal state, a maintenance engineer is dispatched to check the machine state and conduct a fault diagnosis [1]. In order to meet the industry’s demand for maintenance technology, the focus of maintenance has gradually developed into technologies such as condition monitoring, predictive maintenance, and early fault diagnosis [1]. Identifying problems early and predicting the life cycle of a machine for early preventive maintenance or updating equipment can avoid costly plant failures or downtime. Milling cutters are an important part of product processing and are often damaged items on machine tools. This paper uses a milling cutter as the object of the experiment. For the milling cutter of the machine tool, image processing or optical instrument detection is generally used, and the vibration method, radiation sensor, or ultrasonic detection are also used. These methods are useful and even accurate. Compared with other sensing and monitoring instruments, sound sensors are inexpensive, portable, and have less computational data. Therefore, this paper proposes to monitor the wear of milling cutters in machine tools by sound, predict the life cycle of milling cutters, make it preventative protection, and replace and repair equipment before problems occur, thereby reducing product loss and maintaining its accuracy. When recording the milling audio, other noises not related to the milling audio, such as coolers, fans, etc., are also recorded at the same time. In order to avoid the interference of these noises, we developed a noise reduction model applied to milling audio, which can effectively reduce the impact of noise on the life cycle model and obtain more accurate prediction results. Combining these two models, we build a complete life cycle estimation system that can be applied to the operating environment of the machine tool.

The milling process has many details to consider. They may affect life cycle estimation. Eser et al. [2] used milling parameters as input to ANN and used ANN to estimate the surface roughness. Kara et al. [3] tested the effect of hard turning on surface roughness and tool wearing. Regarding the research on life cycle estimation, Li et al. [4] used deep learning and feature extraction to estimate the remaining life of the machine. Chen et al. [5] proposed to use MFCC to capture the features of the punch sound, and then use DNN training to estimate the life cycle of the punch. DNNs are a common practice in speech recognition [6,7,8] and are also commonly used in life cycle prediction models for machines and tools. Boli et al. [9] analyzed the feasibility of applying mask noise reduction in machine tools. Masking is a common denoising method that is often used as a training target for neural networks. Saleem et al. [10] tested the effect of different masks applied to deep neural networks. Wang et al. [11] proposed that an Ideal Ratio Mask (IRM) can be more objective and more suitable as a training target for DNN. However, the IRM does not consider the correlation between noise and audio, so some studies proposed improvements in this part. Xia et al. [12] proposed to use an Optimal Ratio Mask (ORM) as a training objective for supervised learning to achieve better noise reduction. In this paper, based on masks and DNN, using noise addition and feature extraction, a noise reduction model suitable for machine tools is proposed to improve the estimation of the tool life cycle.

The architecture for the remainder of this article is as follows. In Section 2, we illustrate the establishment of the life cycle model. In Section 3, we illustrate the architecture of the noise reduction model. Experimental results and analysis will be presented in Section 4. Section 5 provides conclusions.

## 2. Life Cycle Estimation Methodology

As shown in Figure 1, the life cycle prediction system is divided into several parts. First, we record the milling audio. Second, we extract the features of the audio, for this part we use MFCC. In addition, we calculate the correlation coefficient between features and milling times and compare the correlations to find suitable features as the input data of DNN. Finally, after the DNN training is complete, we test the accuracy of the model.

We recorded milling audio in the environment shown in Figure 2 and used the equipment shown in Figure 3. This paper uses Mini 5-Axis-CNC (Figure 3a) as the recording object. As the shaft rotates, it allows precision machining. An aluminum block (Figure 3b) is the machined material. We cut 30 layers on each block. The milling cutter (Figure 3c) is made of tungsten steel, and the cutting shape is spiral. Its side length is 12 mm, its diameter is 4 mm, and the number of teeth is 3. Our milling object is an aluminum block, which is too soft. Milling parameters must be carefully adjusted to suit the material.

This article started testing different milling methods and milling parameters in December 2019 and completed the setup in June 2020. Cutting information is shown in Table 1. After milling, the wear of the milling cutter is difficult to observe with the naked eye. Milling cutter wear must be seen with a microscope. However, the status of the tool can be quickly confirmed by sound. The iPhone is a portable device with easy access and recording capabilities. Using a professional acoustic emission sensor, the recording effect will be better. We use an iPhone 11 as a recorder with a sampling frequency of 44.1 KHz. A voice recorder is located next to the machine tool and it records clear milling audio. The position of the recorder is fixed each time to avoid deviations between recorded audios.

From September 2020 to May 2021, we recorded milling audio once a week for 5 h. According to the cutting time of the tool, one audio file is recorded every 10 min, and a total of 25 audio files are recorded at a time. We recorded a total of 900 audio files of 5 s each to record the changes in the sound of the tool as the cutting time increased.

There are many algorithms for feature extraction, and MFCC is a more common method [13,14,15,16]. It is a speech feature algorithm developed based on human hearing, which can imitate the features obtained by the human ear in different frequency bands.

Figure 4 is a flow chart of the MFCC. In the pre-emphasis stage, the signal will go through a high-pass filter to compensate for the loss of high-frequency signals during the vocalization process. The signal is then split into frames and multiplied by the Hamming window. The Hamming window can solve the spectrum leakage and reduce the discontinuity between the sound frames.

Equation (1) is the formula of the Hamming window. *N* represents the number of data points in the sound frame, and *n* is the number of data points. Different smoothing effects can be achieved by adjusting the value of α. After testing, the value of α is set to 0.46.
(1)$$W\left(n,\alpha \right)=\left(1-\alpha \right)*\cos(\frac{2\pi n}{N-1})\;,\;0\leq n\leq \left(N-1\right)\;$$

After pre-processing, the signal is converted from the time domain to the frequency domain through Fast Fourier Transform (FFT). The Mel filter will convert frequencies to Mel-frequencies. Mel-frequency represents the sensitivity of the human ear to frequency, and the relationship with frequency is as follows:(2)$$m=2595log_{10}\left(1+0.0014f\right)$$
(3)$$f=700\left(10^{0.00038m}-1\right)$$

Mel filters are implemented in an approximate way to achieve filtering. In the low-frequency band, the filters are densely spaced and have a narrow bandwidth. As the frequency increases, so do the spacing and bandwidth. This simulates the phenomenon that the human ear is more sensitive at low frequencies than at high frequencies.

After the processing of the Mel filter, a set of energy data is obtained and converted into logarithmic energy. MFCC can be obtained after the energy is subjected to discrete cosine transform (DCT). However, current MFCCs only have the static characteristics of speech. In the delta cepstrum stage, we will add dynamic characteristics to the MFCC to enhance the recognition effect.

Gammatone Frequency Cepstral Coefficients (GFCC) and MFCC are very similar, they differ in the filters used. MFCC uses a Mel filter; GFCC uses a Gammatone filter. Gammatone filters are linear filters implemented by impulse responses and are widely used in auditory systems. Its mathematical expression is as follows:(4)$$g\left(t\right)=at^{n-1}e^{-2\pi bt}cos\left(2\pi f_{c}t+\emptyset \right)$$
where *a* represents the output gain, *n* represents the order of the filter, and *b* represents the length of the impulse response, which determines the bandwidth of the wave filter. $f_{c}$ represents the filter center frequency, and ∅ represents the phase angle. Since the human ear is insensitive to phase, the phase is generally set to 0.

PNCC is a feature extraction method for noise processing [17,18]. It carries out data processing before extraction and uses the features of different changing speeds of speech and noise to reduce the interference of noise so that PNCC can describe the features of the target speech more accurately.

After comparing several feature extraction methods, MFCC can better represent the change of tool milling noise corresponding to the number of milling times, so MFCC is finally selected for the life cycle model. Before bringing the features into the DNN training, we use the correlation coefficient to judge the change of the features with the milling times of the tool, find the features with high correlation, and then bring these features into the DNN [19].

Equation (5) is a formula for the correlation coefficient. *N* represents the number of data points. If the MFCC feature (*Y*) becomes larger as the number of millings (*X*) increases, the correlation coefficient (r) will be positive, and vice versa. In order to facilitate identification, this paper will take the absolute value of the correlation coefficient, and then find out the features that will change with the increase of milling times.
(5)$$r=\frac{\sum^{}XY-\frac{\sum^{}X\sum^{}Y}{N}}{\sqrt{\left(\sum^{}X^{2}-\frac{{\left(\sum^{}X\right)}^{2}}{N}\right)\left(\sum^{}Y^{2}-\frac{{\left(\sum^{}Y\right)}^{2}}{N}\right)}}$$

According to the milling layers, we divided the life of the milling cutters into three cycles. The first cycle milled the fewest layers and the third cycle milled the most. We divide the audio into three parts, 1~300 audio files are in the first cycle, 301~600 audio files are in the second cycle, and 601~900 audio files are in the third cycle. After the audio is marked with a label, it will be divided into a training set and a test set. Figure 5 is the architecture diagram of DNN. According to the correlation coefficient, we select some of the MFCC features as the input of the DNN. The DNN has three hidden layers, each with 64 neurons. The output layer has three neurons, corresponding to the three life cycle stages. Audio files will be categorized into one of the cycles.

In addition, training parameters and the activation function affect the training time and training effect [20]. After testing, the batch size is 10 and the epoch is 300. The ReLU activation function works best. We chose it as the activation function for the life cycle model. After the model was trained, we would calculate the accuracy of the life cycle model. As shown in Figure 6, three colored dots represent audios with different life cycles. The range of values for each cycle is 1. From the first cycle to the third cycle, the range of value is 0 to 3. We calculated accuracy using the number of audios classified in the correct cycle and the total number of audios. Compared with other architectures, this DNN architecture has relatively good accuracy.

Table 2 is the correlation between MFCC features and tool loss. First, we judge the 14 features separately and then arrange the features of each sound file according to the order of the milling times of the tool. Finally, we can calculate the correlation between each feature and the number of times the tool has been milled. However, the correlation of some features is very low. In order to avoid features that are not related to the milling times of the tool from affecting the accuracy, the 14 features in this paper are reduced in order from low to high correlation, as shown in Table 3. Finally, it is calculated that the accuracy rate is the highest when 6 features are brought into the tool life cycle model, which can reach 76%. In addition, regarding the ratio of the training set and test set, as shown in Table 4, we tried 90% training and 10% testing, 80% training and 20% testing, and 70% training and 30% testing, and found that 80% training data and 20% testing data has the most accurate data.

## 3. Noise Reduction Methodology

The audio of milling is similar to the audio of noise, and it is most noticeable in the low-frequency part, and the noise cannot be separated using ordinary low-frequency filters [21,22,23]. Moreover, the sound of processing sometimes varies greatly, which affects life cycle estimation. In this paper, mask-based supervised speech separation is used to formulate these problems as a supervised learning problem [24,25]. Figure 7 is a flowchart of the noise reduction model. This paper will first record the milling audio and noise, and then use them to generate the Optimal Ratio Mask (ORM) as the learning target of DNN [26,27]. In addition, this paper extracts the feature, PNCC, from the test signal as the input data of DNN. After DNN training is completed, the degree of noise suppression can be determined by features. Finally, we will use the PESQ and STOI to evaluate the effect of noise reduction.

We used the milling sound recorded without a fan on as the clean audio. Furthermore, we recorded the sound of the cooling fan when the machine tool was not milling as noise and mixed it with the milling sound at 10 db to get a noise-like signal. Finally, we create masks using clean audio and noisy audio. Mask is a commonly used method for noise reduction, which is usually trained by deep learning. First, add the clean signal to the known noise, after the calculation of the mask, and then enter the deep learning training, after obtaining the mask model of the relationship between the target audio and noise, the speech separation or signal enhancement can be achieved.

A signal (*x*) with additive noise (*n*) can be mathematically represented as:(6)y(t)=x(t)+n(t)

After Fourier transforming Equation (6), we get Equation (7):(7)Y(t,f)=X(t,F)+N(t,f)

Each time-frequency unit is called a TF unit. In the frequency domain, the value of each TF unit falls in the complex domain. From the perspective of amplitude, it is necessary to restore |*X*|, and the Mask matrix should be as follows:(8)$$M=\frac{\left|X\right|}{\left|Y\right|}=\frac{\left|X\right|}{\left|X\right|+\left|N\right|}$$

IRM calculates the energy ratio of each TF-unit signal and noise and adjusts the energy of the TF-unit. IRM reflects the degree of noise suppression of each TF unit, which can further improve the quality of the separated audio. The equation of IRM is as follows:(9)$$IRM\left(t,f\right)={\left(\frac{{\left|S\left(t,f\right)\right|}^{2}}{{\left|S\left(t,f\right)\right|}^{2}+{\left|N\left(t,f\right)\right|}^{2}}\right)}^{\beta }$$

The value range of IRM(t,f) is between 0 and 1, and β is usually 0.5. ${\left|S\left(t,f\right)\right|}^{2}$ and ${\left|N\left(t,f\right)\right|}^{2}$ are the signal and noise energy values in the TF unit.

ORM is an improvement of IRM, which calculates the mean square error (MSE) of the target audio S(t,f) and the masked audio $\hat{S}\left(t,f\right)$. Equation (10) represents the transformation between S(t,f) and $\hat{S}\left(t,f\right)$, and γ(t,f) represents ORM. Equation (11) is the calculation of MSE.
(10)$$\hat{S}\left(t,f\right)=\gamma \left(t,f\right)\left[S\left(t,f\right)+N\left(t,f\right)\right]$$
(11)$$L\left(t,f\right)=\frac{1}{T}\sum^{}{\left|\hat{S}\left(t,f\right)-S\left(t,f\right)\right|}^{2}$$

Substituting Equation (10) into Equation (11) can get the following equation:(12)$$L\left(t,f\right)=\frac{1}{T}\left(\sum^{}{\left(\gamma \left(t,f\right)-1\right)}^{2}S{\left(t,f\right)}^{2}+\gamma {\left(t,f\right)}^{2}N{\left(t,f\right)}^{2}+\sum^{}2\gamma \left(t,f\right)\left(\gamma \left(t,f\right)-1\right)ℛ\left(S\left(t,f\right)N^{*}\left(t,f\right)\right)\right)$$

In Equation (12), the superscript * denotes the conjugate operator, and R(∙) returns the real part of the complex number. Partially differentiate L(t,f) with respect to γ(t,f) and let ∂L(t,f)/∂γ(t,f) = 0 to find the minimum MSE, the result is as in (13).
(13)$$\gamma \left(t,f\right)=\frac{{\left|S\left(t,f\right)\right|}^{2}+ℛ\left(S\left(t,f\right)N^{*}\left(t,f\right)\right)}{{\left|S\left(t,f\right)\right|}^{2}+{\left|N\left(t,f\right)\right|}^{2}+2ℛ\left(S\left(t,f\right)N^{*}\left(t,f\right)\right)}$$

In Equation (14), because the range of γ(t,f) value is (−∞, +∞), the hyperbolic tangent function (tanh) formula is added to converge the ORM value, and 𝐾 can be adjusted with the value of 𝑐 to face different noise sources and target audio.
(14)$$ORM\left(t,f\right)=K\frac{1-e^{-c\gamma \left(t,f\right)}}{1+e^{-c\gamma \left(t,f\right)}}$$

ORM is the result of calculating the minimum MSE, and the complex part is also added to the calculation, so compared with IRM, the details are clearer, and the relationship between audio and noise can be subtler.

After feature extraction, the noisy signal will be input into the DNN as the basis for noise reduction. The feature extraction method will affect the effect of the noise reduction system. Figure 8 is the flow chart of PNCC. In the pre-processing stage, the signal will go through FFT and Gammatone Filter Bank to convert it into features similar to the human auditory model.

In the noise suppression stage, PNCC undergoes more complex processing to reduce the impact of noise. In the part of the medium-time power calculation, because the noise energy changes slower than the speech, a larger window can be used to obtain better results. Then, the background noise is suppressed by Asymmetric Noise Suppression, where temporal masking can reduce the influence of reverberation. Spectral weight smoothing enables speech enhancement and noise compensation, while mean power normalization reduces the effects of amplitude scaling. In addition, the energy is processed by an exponential nonlinear function, making it more in line with human auditory perception.

After the noise suppression stage, DCT is used to convert it into acoustic parameters, and then Cepstral Mean Normalization (CMN) is performed on the features to make the distribution between the data more similar, and the PNCC operation is completed. Compared with other feature extraction methods, PNCC handles noise more completely, and it has better effects on audio with noise.

After the signal is denoised, it is necessary to score the denoising quality. However, there is no corresponding database and scoring method for the sound of the machine tool. Therefore, this paper uses the scoring mechanisms of speech, STOI and PESQ, as the scoring method for machine tool noise reduction.

PESQ is a common method for evaluating speech quality. It compares the difference between the original input signal and the processed signal and uses the MOS scoring method to quantify the difference between the ideal speech model and the actual output signal. As shown in Figure 9, the PESQ scoring method corresponds to the MOS score of 4.5 to −0.5, representing the best and worst scores, respectively.

STOI is a measure of speech intelligibility. STOI is scored by comparing the clean speech and the speech to be evaluated, and the value range is 0–1. The higher the value, the better the voice quality.

Table 5 and Table 6 are the comparison of different feature extraction methods and the noise reduction model trained by the mask. It can be found that the combination of PNCC and ORM has the best effect under the PESQ and STOI scores. Finally, we choose PNCC and ORM as the input of DNN.

Figure 10 is the architecture of DNN. Before extracting the signal, the signal will extract the features through the Gammatone Filter and temporarily store them. After feature extraction, IRM and ORM are used as training targets to perform Label actions. If the training target is IRM, assign a value to the Label according to the ratio of the signal to noise, and the range of the Label is between 0 and 1. If the training target is ORM, the value of Label is between −K and K. Label the T-F Unit captured by each feature, so that the supervised learning effect can be more accurate according to the value of the Label during the subsequent DNN training. After the DNN obtains the value of the Label and the audio features, it will use the features to make a noise reduction mask suitable for the domain, which is equivalent to the inverse conversion of the Gammatone Filter, and the noise reduction is completed.

Figure 11 shows the DNN training process of 1 channel. DNN will be trained according to the features of the label and the noise-added signal, and obtain a noise reduction mask suitable for audio. Each DNN corresponds to a frequency band, and 5 sound frames are learned at a time. Combining the DNNs of multiple channels together can achieve a mask suitable for complete audio and complete noise reduction.

Table 7 shows the noise reduction effect of different hidden layers and Gammatone Filter channels. It can be found that 4-layer hidden layers and 64-channel Gammatone Filter will have better training results. Therefore, the noise reduction model finally selects the 64-channel Gammatone Filter to extract features and establishes 4 hidden layers with a total of 1024 neurons. We also tested different parameters for the noise reduction model. We use ReLU as an activation function. The learning rate is 0.001, the epoch is 350 and the batch size is 32.

Table 8 shows the noise reduction models trained with different data ratios. It can be found that the training results of 70% of the training set and 30% of the test set will have a better noise reduction effect.

## 4. Experimental Results and Discussion

It can be seen from Table 9 that the noise reduction system in this paper can effectively reduce the noise whether the noise is larger or the target signal is larger. It is obvious from the scores of STOI and PESQ that the noise reduction system in this paper can reduce the noise when the SNR is −4 dB to 4 dB.

The SNRs of the audios in Table 10 are all −4 db, using PNCC to extract features. It can be found that compared with IRM, ORM has better results as the training target of the noise reduction model. The noise added to the noise, whether it is mechanical sound or human voice, has been significantly improved after the noise reduction system, so the noise reduction system in this article has a good noise reduction effect in different environments.

As shown in Figure 12, before using milling audio for life cycle estimation, the audio will be passed through a noise reduction system to eliminate the influence of noise, and then feature extraction and correlation coefficient analysis will be performed. Table 11 is the comparison of the correlation coefficients before and after noise reduction. Comparing the correlation coefficients between the 14 features and the milling times of the tool, it can be found that m1, m2, m4, m5, m6, m7, m8, m9, and m10 have all improved. This can know that when reducing noise, the change of the sound can be found from the features, and a better effect can be obtained.

As shown in Table 12, the accuracy can be increased to 0.8 when 6 features are selected after noise reduction. It can be seen that the selection of the number of features and the existence of noise will indeed affect the judgment of the tool milling sound in the life cycle model. Therefore, this paper finally chooses to use 6 features as the number of features for judging the life cycle of the tool and uses the noise reduction system to increase the accuracy.

## 5. Conclusions

This paper successfully establishes a machine tool life cycle model based on deep learning and uses the noise reduction model to effectively remove the fan sound contained in the milling sound of the machine tool. Three methods are compared in the life cycle model to capture the sound features of the machine tool. Among them, MFCC has the best performance, and in the noise reduction model, PNCC has the best performance. Under the STOI score, PNCC works well with IRM and ORM. The lift rate reaches 5% and 6.8%, which are higher scores than other methods. PNCC also performs best under the PESQ score. In terms of noise reduction, Mask compares IRM and ORM. Finally, ORM is selected as the target of DNN learning. On the DNN architecture, four hidden layers are selected as the architecture of the entire neural network. It performs better on PESQ and STOI scores than other architectures.

We detect the SNR value of the machine tool site at 10 dB and reduce the noise of tool milling according to this SNR value. The scores of STOI and PESQ have been improved respectively, indicating that this noise reduction model can make noise reduction in line with the ambient sound of the scene. Finally, the noise reduction model is integrated with the tool life cycle model, and noise reduction is carried out according to the actual SNR value of the machine tool site of 10 dB. In this way, the accuracy of the tool life cycle model is increased from 76% to 80%.

There are many different noises in the operation of machine tools. Increasing the types of noise can improve the noise reduction model. Also, when changing the milling parameters, the milling sound also changes. Current lifecycle models test relatively few processing parameters. Testing more parameters may increase the accuracy of life cycle estimation and bring it closer to industrial applications. Finally, the iPhone is not a dedicated recording device. Recording equipment can use industry-standard acoustic emission sensors. The recording effect will then be better, and the life cycle model would be more accurate.

## Figures and Tables

**Figure 1 sensors-23-00284-f001:**
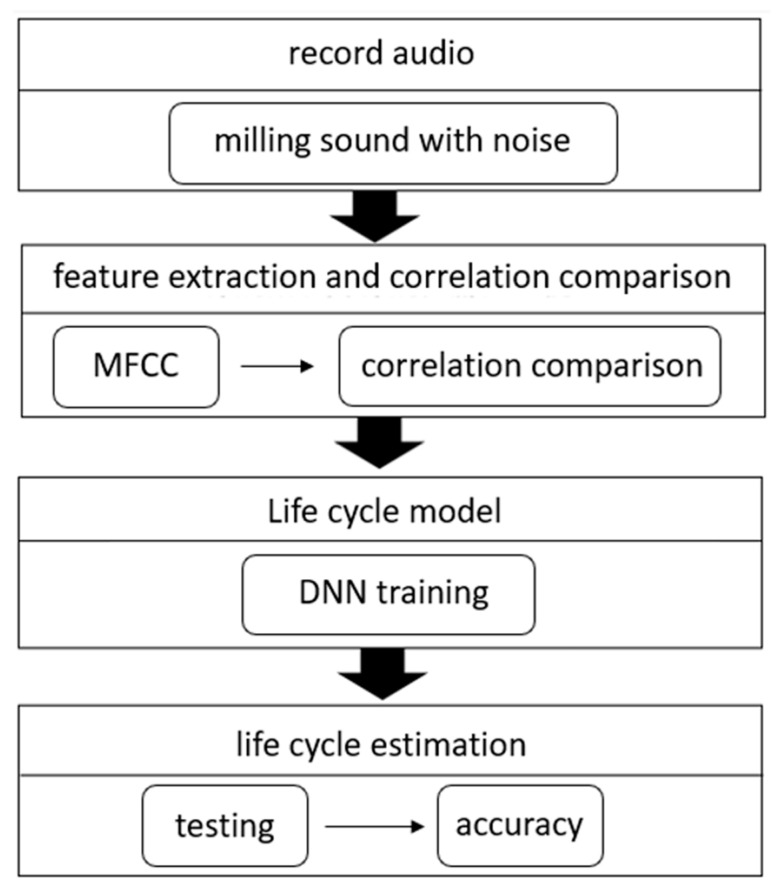
Flow chart of life cycle prediction system.

**Figure 2 sensors-23-00284-f002:**
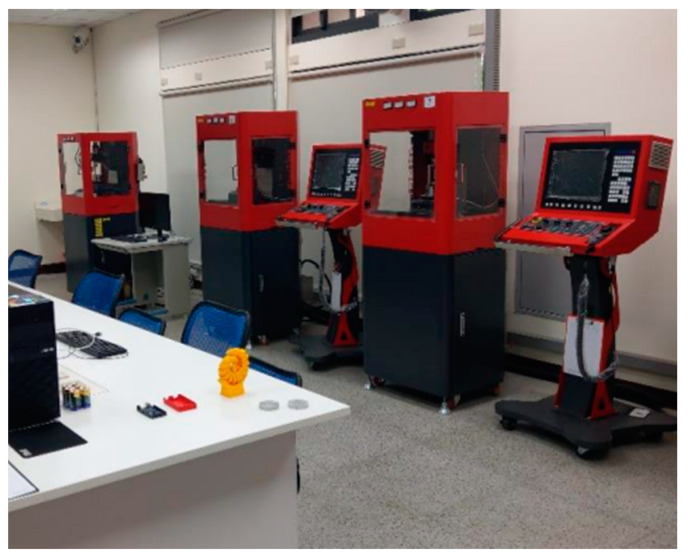
Audio recording environment.

**Figure 3 sensors-23-00284-f003:**
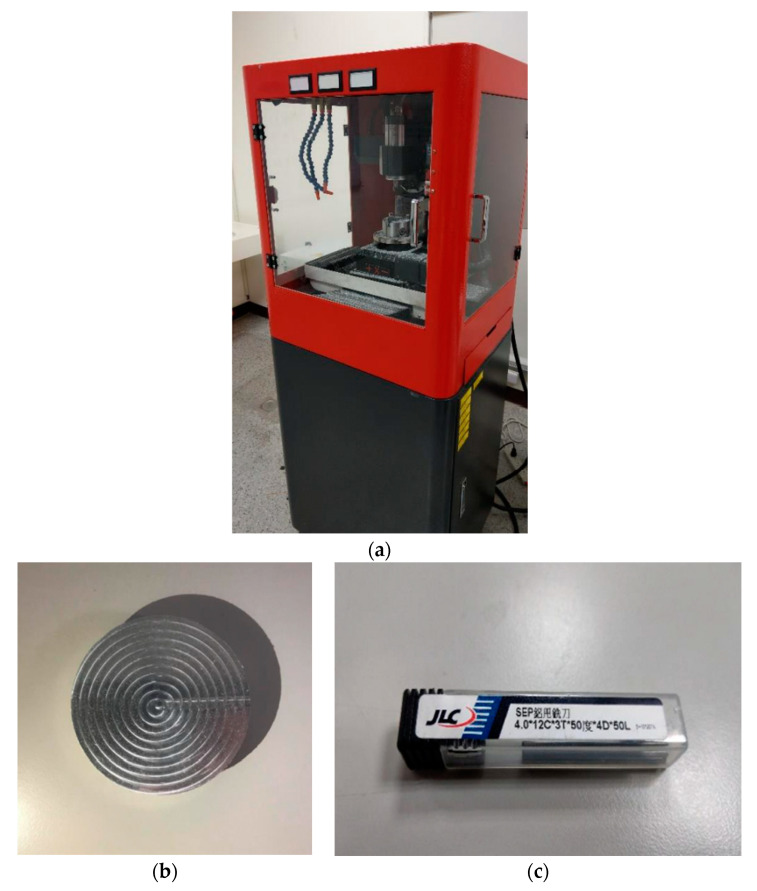
Machine tools and equipment. (**a**) Mini 5-Axis-CNC small five-axis machine tool; (**b**) aluminum for cutting; (**c**) milling cutter for cutting.

**Figure 4 sensors-23-00284-f004:**
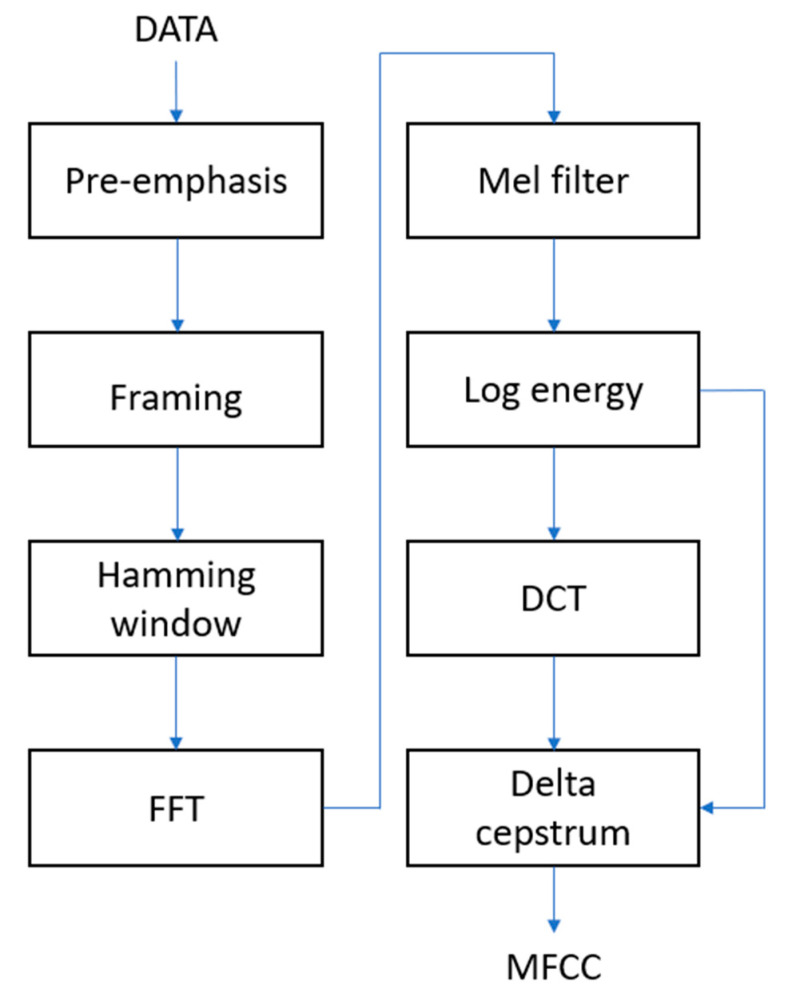
Flow chart of MFCC.

**Figure 5 sensors-23-00284-f005:**
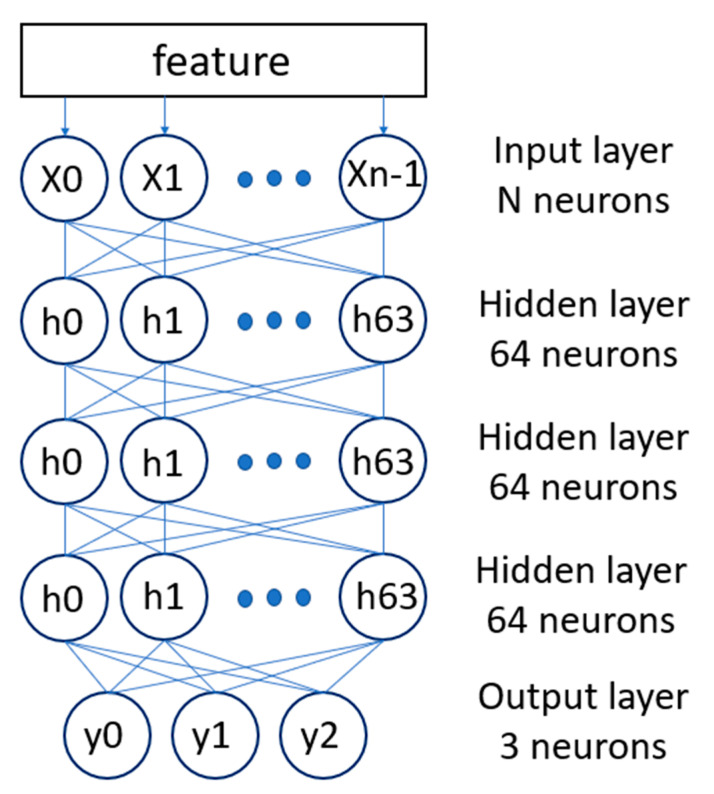
DNN Architecture of Life cycle Model.

**Figure 6 sensors-23-00284-f006:**
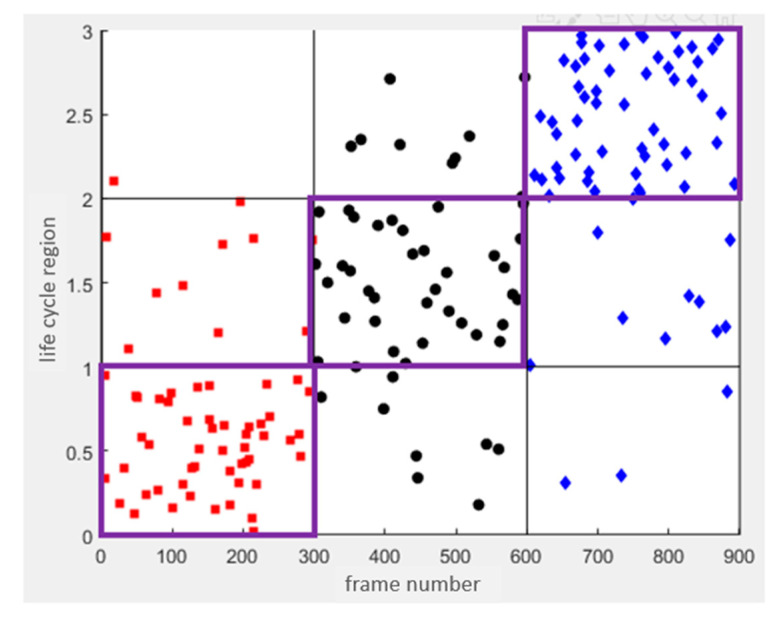
Testing Result of Life Cycle Model.

**Figure 7 sensors-23-00284-f007:**
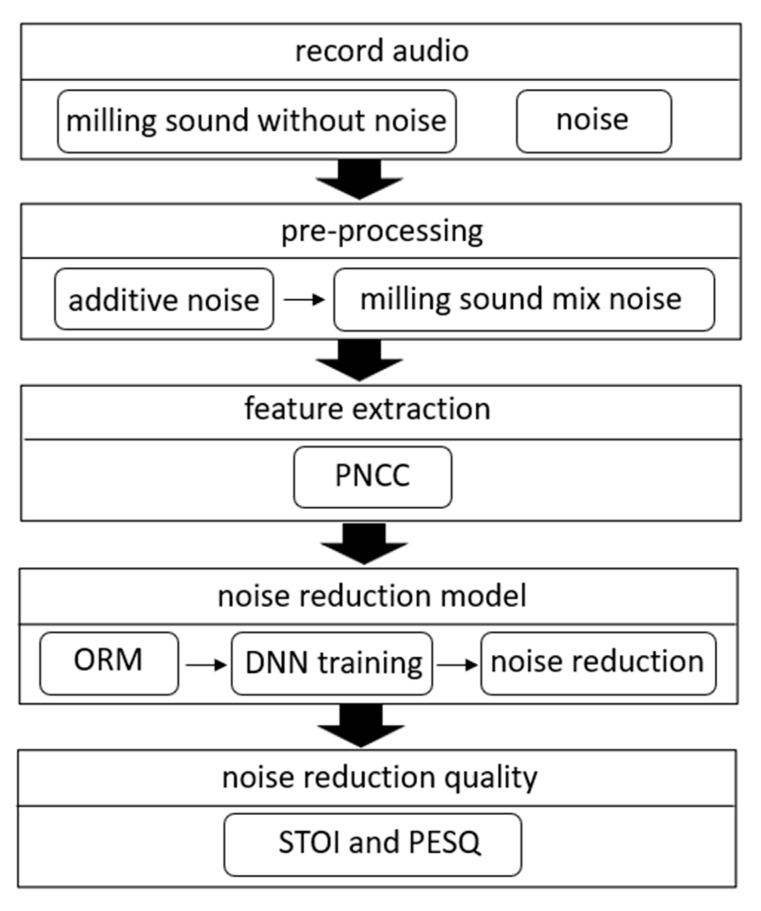
Flow chart of noise reduction model.

**Figure 8 sensors-23-00284-f008:**
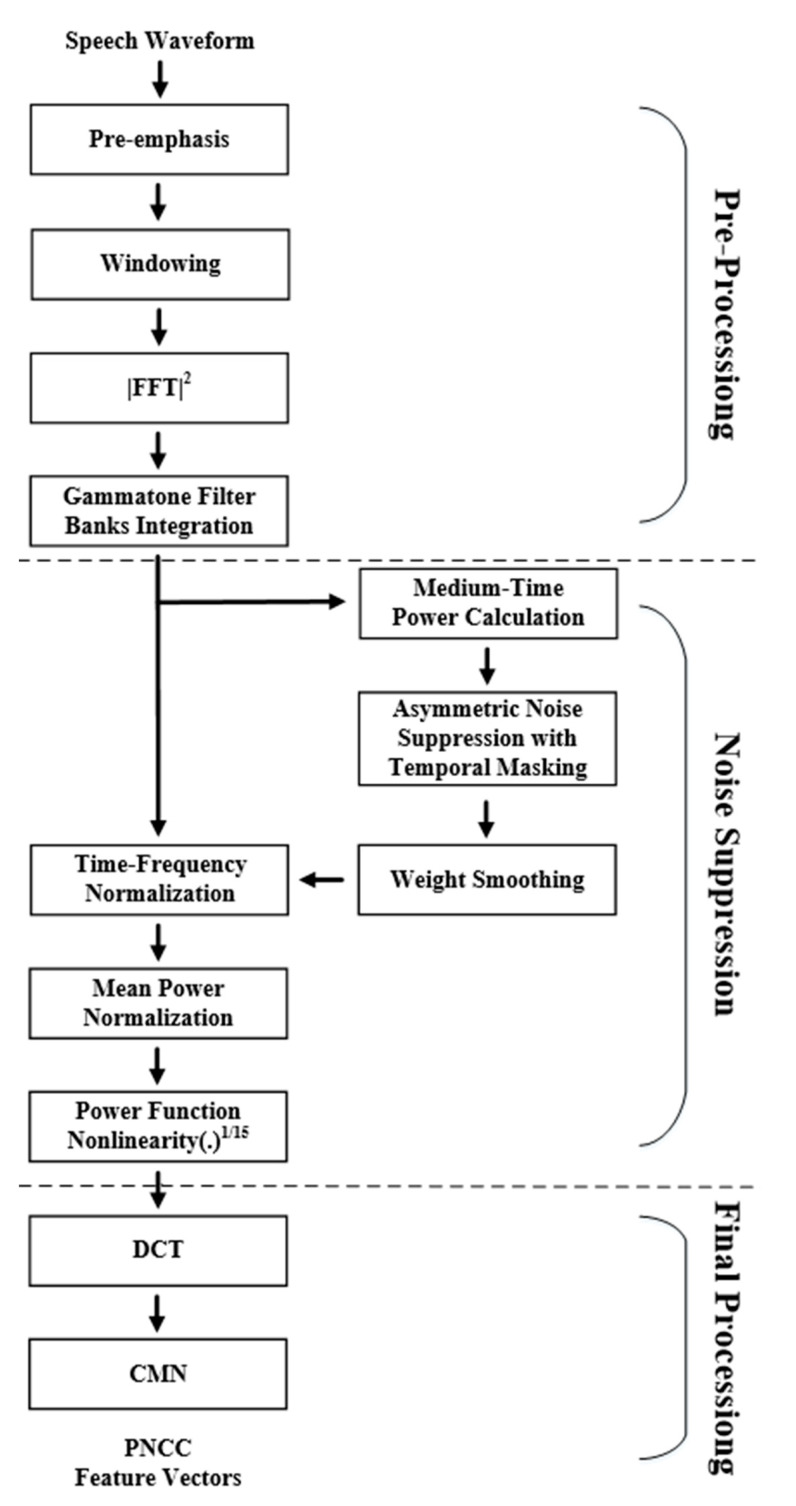
Flow chart of PNCC.

**Figure 9 sensors-23-00284-f009:**
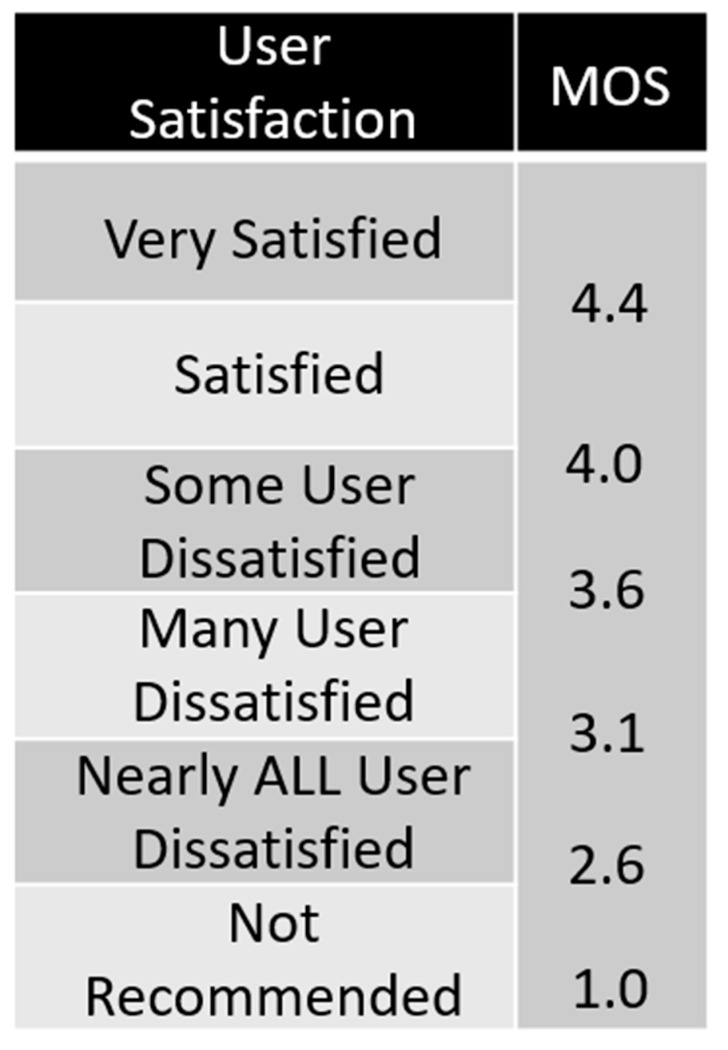
PESQ Quality Chart.

**Figure 10 sensors-23-00284-f010:**
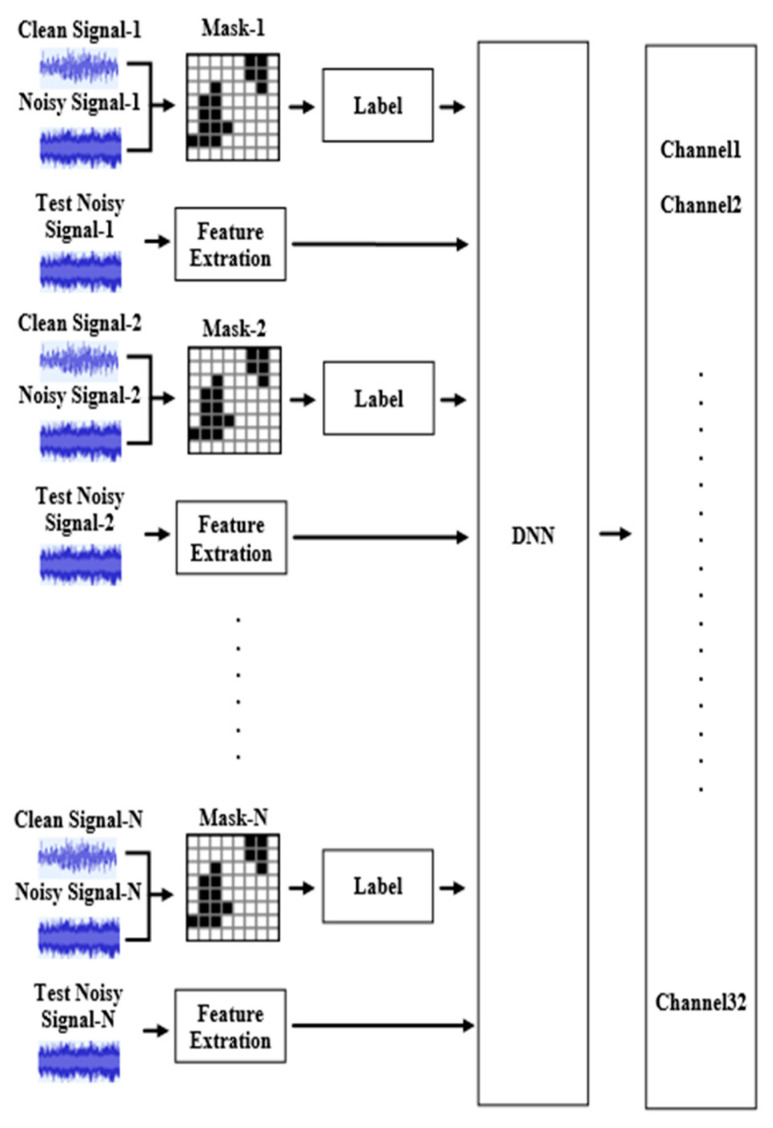
DNN training Mask architecture diagram.

**Figure 11 sensors-23-00284-f011:**
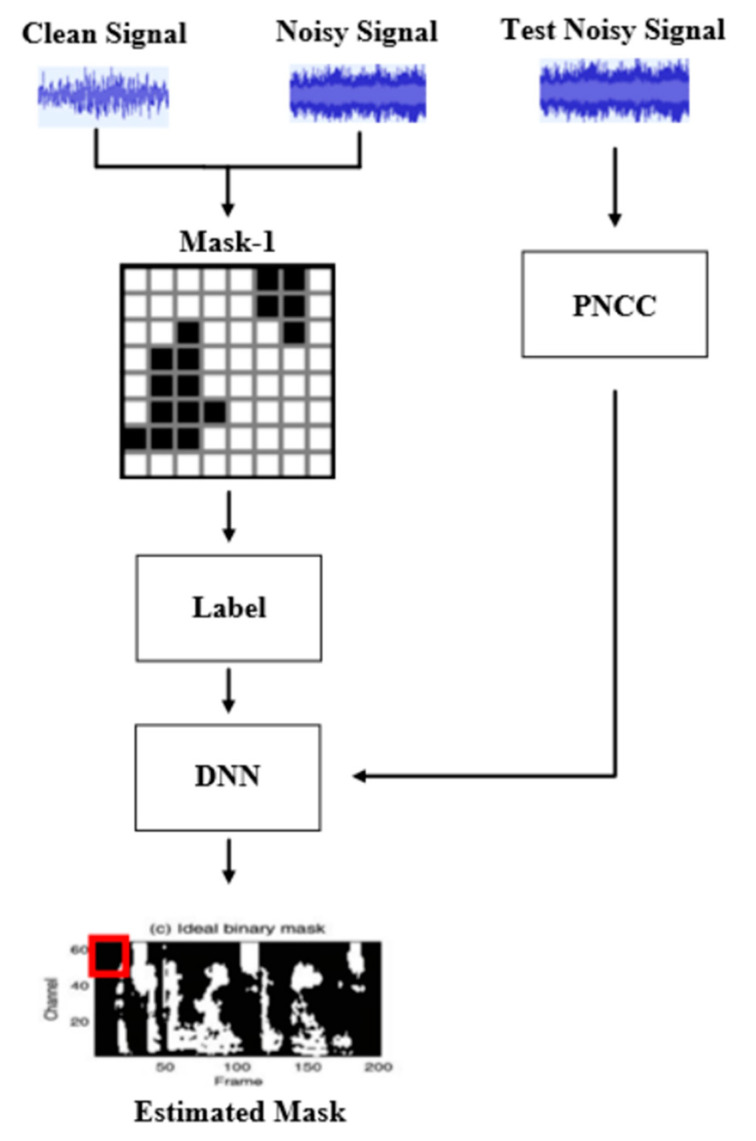
DNN trained per channel.

**Figure 12 sensors-23-00284-f012:**
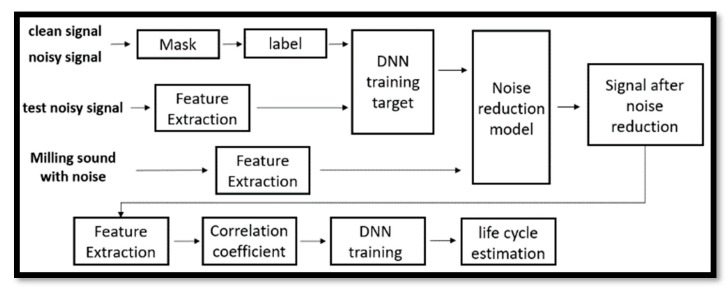
Flow chart of life cycle estimation system after noise reduction.

**Table 1 sensors-23-00284-t001:** Cutting information.

Information	Description
Cutting method	Milling
Processing material	aluminum block
Cutting tool	4 mm milling cutter for aluminum
Processing conditions	Dry
Feed rate	500 mm/min
Cutting depth	0.1 mm
Motor revolution	20,000 times/min
Cutting shape	Circle
Tool moving speed	30 mm/s

**Table 2 sensors-23-00284-t002:** Correlation between MFCC features and the number of milling layers.

Features MFCC	Original File
m1	0.5061
m2	0.3674
m3	0.5065
m4	−0.6690
m5	0.6595
m6	−0.7395
m7	0.8560
m8	0.6102
m9	0.7605
m10	−0.1400
m11	−0.7142
m12	−0.5461
m13	0.3731
m14	0.7723

**Table 3 sensors-23-00284-t003:** Tool life cycle model accuracy.

Before Noise Reduction	
Number of Features	Accuracy
1	52%
2	56%
3	58%
4	66%
5	72%
6	76%
7	75%
8	72%
9	68%
10	68%
11	58%
12	55%
13	55%
14	57%

**Table 4 sensors-23-00284-t004:** Model accuracy under different training data.

Test Data Percentage	Accuracy
Test data (90%)	75%
Training materials (10%)	
Test data (80%)	76%
Training materials (20%)	
Test data (70%)	73%
Training materials (30%)	

**Table 5 sensors-23-00284-t005:** Feature Extraction Method and Mask Comparison 1.

**PESQ**	**MFCC**	
**MASK**	**IRM**	**ORM**
Noise decibel	10 dB	10 dB
before	3.76	3.76
after	3.82	3.84
promote	0.06	0.08
lift rate	1.6%	2.1%
**PESQ**	**PNCC**	
**MASK**	**IRM**	**ORM**
Noise decibel	10 dB	10 dB
before	3.76	3.76
after	3.87	3.91
promote	0.11	0.15
lift rate	2.9%	3.9%
**PESQ**	**GFCC**	
**MASK**	**IRM**	**ORM**
Noise decibel	10 dB	10 dB
before	3.76	3.76
after	3.87	3.86
promote	0.11	0.1
lift rate	2.9%	2.6%

**Table 6 sensors-23-00284-t006:** Feature Extraction Method and Mask Comparison 2.

**STOI**	**MFCC**	
**MASK**	**IRM**	**ORM**
Noise decibel	10 dB	10 dB
before	0.7963	0.7963
after	0.8351	0.8496
promote	0.0388	0.0536
lift rate	4.8%	2.1%
**STOI**	**PNCC**	
**MASK**	**IRM**	**ORM**
Noise decibel	10 dB	10 dB
before	0.7963	0.7963
after	0.8362	0.8506
promote	0.0399	0.0543
lift rate	5%	6.8%
**STOI**	**GFCC**	
**MASK**	**IRM**	**ORM**
Noise decibel	10 dB	10 dB
before	0.7963	0.7963
after	0.8362	0.8492
promote	0.0399	0.0529
lift rate	5%	6.6%

**Table 7 sensors-23-00284-t007:** Scoring of DNN Layer and Channel Selection.

STOI	3 Layers		
Aisle	32	64	128
Promote	0.0187	0.0193	0.0193
**PESQ**	**3 layers**		
Aisle	32	64	128
Promote	0.07	0.12	0.11
**STOI**	**4 layers**		
Aisle	32	64	128
Promote	0.0533	0.0543	0.0543
**PESQ**	**4 layers**		
Aisle	32	64	128
Promote	0.09	0.15	0.12
**STOI**	**5 layers**		
Aisle	32	64	128
Promote	0.0549	0.0519	0.234
**PESQ**	**5 layers**		
Aisle	32	64	128
Promote	0.07	0.09	0.09

**Table 8 sensors-23-00284-t008:** Test data ratio selection.

**STOI**	**PNCC_DNN_ORM**	
Ratio	Training (80%) Testing (20%)	Training (70%) Testing (30%)
SNR (dB)	10	10
Before	0.7963	0.7963
After	0.8327	0.8506
Promote	0.0364	0.0543
Lift rate	4.5%	6.8%
**PESQ**	**PNCC_DNN_ORM**	
Ratio	Training (80%) Testing (20%)	Training (70%) Testing (30%)
SNR (dB)	10	10
Before	3.76	3.76
After	3.89	3.91
Promote	0.13	0.15
Lift rate	3.4%	4%

**Table 9 sensors-23-00284-t009:** Noise reduction effect of different noises.

**STOI**	**PNCC_DNN_ORM**		
Noise source	Cooling fan		
SNR (dB)	−4	0	4
Before	0.1948	0.3271	0.4873
After	0.5094	0.5895	0.6810
Promote	0.3146	0.2642	0.1937
Lift rate	162%	80%	40%
**PESQ**	**PNCC_DNN_ORM**		
Noise source	Cooling fan		
SNR (dB)	−4	0	4
Before	1.312	1.495	1.189
After	2.233	2.233	2.861
Promote	0.920	0.73	1.042
Lift rate	70%	49%	57%

**Table 10 sensors-23-00284-t010:** Different ratio of noise reduction effect.

Machine Tool Cutting Sound (Noise Decibel −4 dB)_PNCC				
**Noise Source**	**Cooler**		**Factory Noise**	
Mask	IRM	ORM	IRM	ORM
Before	1.512	1.512	1.233	1.233
After	2.499	2.805	2.466	2.820
Promote	0.987	1.293	1.233	1.587
Lift rate	65%	86%	100%	129%
**Noise source**	**Human voice**		**White noise**	
Mask	IRM	ORM	IRM	ORM
Before	1.145	1.145	1.421	1.421
After	2.558	2.835	2.456	2.807
Promote	1.413	1.690	1.035	1.386
Lift rate	123%	148%	73%	98%

**Table 11 sensors-23-00284-t011:** Comparison of correlation coefficients between MFCC and milling times after noise reduction.

Features MFCC	Original File	After Noise Reduction	Lift Rate
m1	0.5061	0.5633	11%
m2	0.3674	0.8316	126%
m3	0.5065	0.3541	−15%
m4	−0.6690	−0.7334	9%
m5	0.6595	0.6913	5%
m6	−0.7395	−0.7592	3%
m7	0.8560	0.8613	0.5%
m8	0.6102	0.6888	12%
m9	0.7605	0.7721	1.5%
m10	−0.1400	−0.1717	3%
m11	−0.7142	−0.2020	−71%
m12	−0.5461	−0.1635	−70%
m13	0.3731	0.3650	−2%
m14	0.7723	0.4494	−41%

**Table 12 sensors-23-00284-t012:** The accuracy of the tool life cycle model after noise reduction.

Before Noise Reduction		After Noise Reduction		Lift Rate
Number of Features	Accuracy	Number of Features	Accuracy	
1	52%	1	61%	17%
2	56%	2	61%	9%
3	58%	3	64%	10%
4	66%	4	68%	3%
5	72%	5	77%	7%
6	76%	6	80%	5%
7	75%	7	79%	5%
8	72%	8	77%	7%
9	68%	9	77%	13%
10	68%	10	72%	6%
11	58%	11	68%	17%
12	55%	12	68%	24%
13	55%	13	64%	16%
14	57%	14	66%	16%

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
