# Peer review of "Design and Implementation of Machine Tool Life Inspection System Based on Sound Sensing"

_sensors, 2022, doi:10.3390/s23010284_

Round 1

Reviewer 1 Report

The paper presents a  practical and efficient  machine tool life cycle model with noise reduction based on sound sensing. The work has industrial relevance, but the following factors need to be addressed for publication:

1. Milling tests need to be described in more detail. A specific definition of tool life cycle should be given. It is recommended to list the picture information of tool wear.

2. For the tool life model, please indicate whether the tool life, tool wear value or other parameters are the output of the model.

3. Acoustic emission sensor (AE) is a common sound monitoring instrument in machining. iPhone 11 is not a standard testing equipment, test parameters and range of measurement can not be adjusted. Please provide proof that the iphone is suitable and how to locate the iphone.

4. Noise reduction model is crucial for sound detection. The machine tool contains many auxiliary devices, which can generate many extra sound factors during operation. How to define clean audio?

5.The analysis of the influence of different machining parameters on the model effect is lacking.

6. The acquisition method of accuracy of the tool life cycle model  is not described.

7.  The cutting parameters change when the parts are machined. Especially, when the parts are cut in and out, the sound changes greatly.How to avoid its effects.

8. The accuracy rate of 0.8 is not suitable for practical application, so it is suggested to list the future research plan in the conclusion.

Author Response

Thank you for your valuable advice.

  1. Milling tests need to be described in more detail. A specific definition of tool life cycle should be given. It is recommended to list the picture information of tool wear.
    • We have supplemented milling testing information in lines 81 to 96. In addition, we add a description of life cycle into the paper in lines 172 to 173 of the paper. The wear of the milling cutter cannot be seen with the naked eye. Must use a microscope to see. Later, due to the relocation of the laboratory, the knives were lost. Sorry no photos of the knives are available.
  2. For the tool life model, please indicate whether the tool life, tool wear value or other parameters are the output of the model.
    • We divide the life of a milling cutter into three cycles. The tool life model classifies the audio into one of the cycles, and this is the output of the model. Relevant descriptions are in lines 178 to 180.
  3. Acoustic emission sensor (AE) is a common sound monitoring instrument in machining. iPhone 11 is not a standard testing equipment, test parameters and range of measurement can not be adjusted. Please provide proof that the iphone is suitable and how to locate the iphone.
    • The iPhone really isn't a standard test device. So we fixed the iPhone's position and volume for each test. In order to achieve consistent standards for each recording. iPhone is portable and easy to measure audio. Relevant descriptions are in lines 92 to 96.
  4. Noise reduction model is crucial for sound detection. The machine tool contains many auxiliary devices, which can generate many extra sound factors during operation. How to define clean audio?
    • We used the milling sound recorded without fan on as the clean signal. Besides, we recorded the sound of the cooling fan when the machine tool was not milling as noise. Relevant descriptions are in lines 227 to 230.
  5. The analysis of the influence of different machining parameters on the model effect is lacking.
    • The machine learning method is the rule of thumb. We fix the parameters like cutting depth and feed rate. The purpose is to reduce the impact of parameter changes and to test that denoising method can improve accuracy. The analysis of the effect of different processing parameters on the model will be considered. Relevant descriptions are in lines 427 to 429.
  6. The acquisition method of accuracy of the tool life cycle model is not described.
    • We explain the calculation method of the accuracy in lines 184 to 188.
  7. The cutting parameters change when the parts are machined. Especially, when the parts are cut in and out, the sound changes greatly. How to avoid its effects.
    • We use noise reduction model to handle the problem of sharply changing sounds. Relevant descriptions are in lines 215 to 217.
  8. The accuracy rate of 0.8 is not suitable for practical application, so it is suggested to list the future research plan in the conclusion.
    • We add improvement directions. Relevant descriptions are in lines 423 to 429.

Reviewer 2 Report

This paper performed to Design and Implementation of Machine Tool Life Inspection System Based on Sound Sensing. The article is, in general, well written but there are some issues that authors should consider to revise in order to improve its quality. Some comments were done in this way:

·         Abstract should be expanded sentences related to the results. The results of the study should be given as numerical percentages.

·         The paper should be also supported by a literature search including relevant and recent papers. Only the most relevant and up-to-date articles on the study should be given. The following recent articles related to optimization may be cited.

https://doi.org/10.1155/2021/5576600.

https://doi.org/10.1016/j.jmrt.2019.11.037.

·         Give the units in mm. For example table 1 "Cutting depth" and "Tool moving speed"

·         Also give the experimental results with graphs. Also give with a table.

·         Let's fix grammatical errors throughout the article.

·         The article should be edited completely according to the journal writing guide.

·         Fractions should be given with dots throughout the article, including figures and tables.

·         Give the characteristics of the cutting tool in more detail. Specify the type of coating. Give the coating thickness and coating material. Also CVD? Is it PVD?

·         Training and test data should be given with a table.

·         What are the learning rate and other parameters?

·         Which activation function was used in the ANN model?

·         Analyzes should be made for different activation functions. Of these, the function that gives the highest MSE should be preferred. The following work can be viewed and cited in this study.

https://doi.org/10.1007/s10443-012-9286-3.

·         Conclusions should be written in more detail adding numeric data.

Author Response

Thank you for your valuable advice.

  • Abstract should be expanded sentences related to the results. The results of the study should be given as numerical percentages.
    • Abstract has been modified. The representation of the accuracy in the paper has been changed to a percentage form. Table 3, Table 4, Table 12 and line 422.
  • The paper should be also supported by a literature search including relevant and recent papers. Only the most relevant and up-to-date articles on the study should be given. The following recent articles related to optimization may be cited.

https://doi.org/10.1155/2021/5576600.

https://doi.org/10.1016/j.jmrt.2019.11.037.

    • We have cited the papers. The relevant descriptions are in lines 52 to 55.
  • Give the units in mm. For example table 1 "Cutting depth" and "Tool moving speed"
    • We change the unit to mm in Table 1.
  • Also give the experimental results with graphs. Also give with a table.
    • We give the experimental results with table. Relevant part is in Table 12.
  • Let's fix grammatical errors throughout the article.
    • We have fixed some grammatical errors.
  • The article should be edited completely according to the journal writing guide.
    • We modify the format according to the template of Sensors.
  • Fractions should be given with dots throughout the article, including figures and tables.
    • We change fractions to decimals in Equation 2 and Equation 3.
  • Give the characteristics of the cutting tool in more detail. Specify the type of coating. Give the coating thickness and coating material. Also CVD? Is it PVD?
    • We add the information of milling cutter in lines 85 to 86. However, the manufacturer doesn’t provide information of coating.
  • Training and test data should be given with a table.
    • These data are recorded in Table 2 and Table 4.
  • What are the learning rate and other parameters?
    • We supplement these materials in lines 182 and lines 344 to 345.
  • Which activation function was used in the ANN model?
    • We use ReLU as the activation function. Relevant descriptions are in lines 182 to 183 and line 344.
  • Analyzes should be made for different activation functions. Of these, the function that gives the highest MSE should be preferred. The following work can be viewed and cited in this study.

https://doi.org/10.1007/s10443-012-9286-3.

    • We have cited and referred to this paper. Relevant descriptions are in lines 181 to 182.
  • Conclusions should be written in more detail adding numeric data.
    • We add some data and illustrations in the conclusion.

Round 2

Reviewer 1 Report

The authors have made changes according to my suggestions. Thus, I recommend for acceptance.

It is suggested to add images of tool wear at different stages and the corresponding sound signals.

Reviewer 2 Report

The authors have made all the corrections previously given. The article can be accepted for publication as it is. Best regards.